# Downstream retraction of preprinted research in the life and medical sciences

**Michele Avissar-Whiting** *

Research Square Company, Durham, North Carolina, United States of America

* michavissar@gmail.com

## Abstract

Retractions have been on the rise in the life and clinical sciences in the last decade, likely due to both broader accessibility of published scientific research and increased vigilance on the part of publishers. In this same period, there has been a greater than ten-fold increase in the posting of preprints by researchers in these fields. While this development has significantly accelerated the rate of research dissemination and has benefited early-career researchers eager to show productivity, it has also introduced challenges with respect to provenance tracking, version linking, and, ultimately, back-propagation of events such as corrigenda, expressions of concern, and retractions that occur on the journal-published version. The aim of this study was to understand the extent of this problem among preprint servers that routinely link their preprints to the corollary versions published in journals. To present a snapshot of the current state of downstream retractions of articles preprinted in three large preprint servers (Research Square, bioRxiv, and medRxiv), the DOIs of the journal-published versions linked to preprints were matched to entries in the Retraction Watch database. A total of 30 retractions were identified, representing only 0.01% of all content posted on these servers. Of these, 11 retractions were clearly noted by the preprint servers; however, the existence of a preprint was only acknowledged by the retracting journal in one case. The time from publication to retraction averaged 278 days, notably lower than the average for articles overall (839 days). In 70% of cases, retractions downstream of preprints were due–at least in part–to ethical or procedural misconduct. In 63% of cases, the nature of the retraction suggested that the conclusions were no longer reliable. Over time, the lack of propagation of critical information across the publication life cycle will pose a threat to the scholarly record and to scientific integrity. It is incumbent on preprint servers, publishers, and the systems that connect them to address these issues before their scale becomes untenable.

## Introduction

The use of preprints as a mode of rapidly sharing research findings in the biological and medical sciences has become ubiquitous over the last decade, and their adoption has particularly surged since the onset of the COVID-19 pandemic in early 2020 [1]. The global public health

**Data Availability Statement:** The Retraction Watch Database is available from Retraction Watch, and requests for this data should be sent to team@retractionwatch.com.

**Funding:** The author(s) received no specific funding for this work.

**Competing interests:** Michele Avissar-Whiting is the Editor in Chief of the Research Square preprint platform. This does not alter my adherence to PLOS ONE policies on sharing data and materials.

emergency drove researchers to deposit preprints as they awaited peer review in a journal, a practice that became widely embraced by major publishers as the pandemic intensified [2]. But this embrace was generally not accompanied by the development of new mechanisms to link the eventual journal publications consistently and unambiguously to previous versions of the work in preprint servers.

In the life sciences, the version of the work that is typically recognized for career advancement purposes and, reportedly, preferred by researchers [3] is the one published in a journal following peer review–often referred to as the version of record. The preprint version, however, does not become inconsequential once a later version of it is published. Preprints are permanent contributions to the scholarly record. They are fully open access and have their own DOIs, so they can be circulated and cited widely long before a later version is published by a journal. Throughout the pandemic, it has not been unusual to see a preprint cited by another preprint within days of its appearance online–an unprecedented pace of collaborative problem-solving.

Often, preprints continue to be cited even once a later version of the work is published in a journal [4]. There are numerous potential reasons for this, ranging from the technical limitations of reference management software (programs do not automatically update preprints with versions of record) to an active choice by the citing researcher. But perhaps the most critical reason is that linking of preprints to associated versions of record has been unreliable at best and often nonexistent at worst. Because they tend to operate on limited budgets, most preprint servers do not have automated mechanisms for updating preprints with links to their journal-published versions [5]. The mechanisms that do exist on a minority of servers have limited fidelity, as links are typically based on fuzzy matching of titles and author lists, which are subject to change.

The issue of nonexistent or unreliable linking becomes particularly salient in instances where a journal-published version of a preprint is retracted or earns an expression of concern, a note typically issued by an editor to alert readers about potential problems with the article. The incidence of retraction, once an extremely rare occurrence, has increased dramatically since the year 2000 [6]. Retractions are typically carried out because a critical issue comes to light, invalidating the results and conclusions of the work. Misconduct is found to be a factor in about half of these cases [7]. However, because it is unusual for journals to acknowledge the existence of a preprint associated with an article, information about a retraction does not reach the preprint server unless the author updates the server or there are procedures in place at the preprint server to explicitly search for such information. There is some data to suggest that there are generally few meaningful differences between preprints and their corollary journal-published articles [8]. Thus, problems discovered in the latter are likely to impact the former. Moreover, the issue of persistent citation of retracted research in the literature [9] will only be exacerbated by a failure to link versions.

Because preprints have only become popular among life and medical scientists in the past few years and research on retractions in general is sparse, there is little information to be found about the intersection of these two domains. In this analysis, I assess linked journal articles from three major life science and clinical preprint servers to present a snapshot of 1) the incidence of retractions among previously preprinted work, 2) the degree to which these events are acknowledged on the preprints, 3) and other characteristics of these retractions.

## Methods

The Retraction Watch database was chosen for this analysis, as it is the most comprehensive database of retractions, errata, and corrections available, containing (at the time of access on

23 November 2021) 31,492 records. The three preprint servers Research Square, bioRxiv, and medRxiv were used in the analysis because they are the largest life and medical science servers with automated mechanisms in place to link to journal-published versions and from which the data are easily retrievable via API. Data from Research Square was accessed directly, and data from bioRxiv and medRxiv were obtained via their APIs (https://api.biorxiv.org/ and https://api.medrxiv.org/, respectively). The DOIs of journal-published versions of the preprints were matched to entries in the Retraction Watch database, and corollary information about the retractions was collected.

Misconduct in each case of retraction was categorized according to the areas defined by the Council of Science Editors [10] as follows: *Mistreatment of research subjects* (includes failure to obtain approval from an ethical review board or consent from human subjects before conducting the study and failure to follow an approved protocol); *Falsification and fabrication of data*; *Piracy and plagiarism* (including unauthorized use of third-party data or other breach of policy by authors); or *No evidence of misconduct*. The determination of the presence and type of misconduct was based on information contained in the individual retraction notices as well as on the reasons for retraction briefly noted in the Retraction Watch database.

To account for the relative recency of preprints, the calculation of the average time-to-retraction for entries in the Retraction Watch database was limited only to articles published after 2014, when the first life science preprints began to emerge on bioRxiv.

## Results

Because the discovery of retractions downstream of preprints relies heavily on the existence of a linkage between the preprint and its journal-published version, I first assessed the proportion of preprints for which such links appear for all three servers. Of all posted preprints on Research Square, bioRxiv, and medRxiv, 24%, 54%, and 35% are linked to a published journal article, respectively (Table 1). When using a 2-year cut-off (i.e., limiting the analysis only to articles posted as preprints more than 2 years ago, as previously done to approximate "publication success" [11]), the values increase to 45%, 73%, and 74% (Table 1).

Among the three preprint servers, a total of 30 downstream retractions were identified: 17 at Research Square, 11 at bioRxiv, and 2 at medRxiv (Table 2). This represents 0.05%, 0.01%, and 0.02% of all preprints with journal publication links at Research Square, bioRxiv, and medRxiv, respectively (Table 1). All 30 of the retracted papers in this analysis had been published in open access journals.

The time from preprint posting to publication in a journal ranged from 41 to 468 days, with an average time of 169 days. The time from publication in the journal to retraction ranged from 11 days to 993 days and averaged 278 days. Among all retractions in the Retraction Watch database for papers published after 2014 (the year that bioRxiv preprints began to appear), the average time to retraction was 678 days.

**Table 1. Summary of preprint volume, occurrence of journal article links, downstream retraction incidence at three preprint servers.**

| | Preprints posted[1] | Preprints with article links (%) | Preprints with article links using 2-year cutoff (%) | Average days from preprint posted to published at journal | Average days from published to retracted at journal | Retractions among linked published articles (% of all preprints with article links) |
|---|---|---|---|---|---|---|
| **Research Square** | 122700 | 29608 (24%) | 4112 (45%) | 130 | 278 | 16 (0.05%) |
| **bioRxiv** | 140683 | 75446 (54%) | 45066 (73%) | 199 | 402 | 11 (0.01%) |
| **medRxiv** | 26458 | 9187 (35%) | 444 (74%) | 179 | 154 | 2 (0.02%) |

[1]all time; excludes versions.

**Table 2. Details for 30 instances of retractions performed on articles with earlier versions posted on preprint servers.**

| | Time from preprint to publication (days) | Time from publication to retraction (days) | Paywalled article | Reason for Retraction | Misconduct classification | Acknowledgment of retraction on preprint | Preprint withdrawn |
|---|---|---|---|---|---|---|---|
| Case 1 | 84 | 372 | No | Concerns/Issues; Duplication of Article; +Unreliable Results; | Mistreatment of research subjects | no | no |
| Case 2 | 165 | 522 | No | Duplication of Article; | Piracy and plagiarism | no | no |
| Case 3 | 188 | 239 | No | Concerns/Issues About Data;+Lack of IRB/IACUC Approval;+Unreliable Data; | Mistreatment of research subjects | no | no |
| Case 4 | 54 | 395 | No | Lack of IRB/IACUC Approval; | Mistreatment of research subjects | no | no |
| Case 5 | 104 | 392 | No | Lack of IRB/IACUC Approval; | Mistreatment of research subjects | no | no |
| Case 6 | 46 | 401 | No | Lack of IRB/IACUC Approval; | Mistreatment of research subjects | no | no |
| Case 7 | 395 | 84 | No | Concerns/Issues About Data;+Conflict of Interest;+Duplication of Image; +Unreliable Results;+Upgrade/Update of Prior Notice; | Falsification and fabrication of data | yes | yes |
| Case 8 | 152 | 454 | No | Author Unresponsive;+Duplication of Image;+Unreliable Data; | Falsification and fabrication of data | no | no |
| Case 9 | 41 | 484 | No | Duplication of Image;+Investigation by Third Party;+Unreliable Results; | Falsification and fabrication of data, Piracy and plagiarism | no | no |
| Case 10 | 104 | 324 | No | Error in Data;+Unreliable Data; +Unreliable Results; | No misconduct | no | no |
| Case 11 | 171 | 278 | No | Duplication of Image;+Unreliable Data; | Falsification and fabrication of data | yes | yes |
| Case 12 | 76 | 183 | No | Duplication of Image;+Unreliable Data; | Falsification and fabrication of data | no | no |
| Case 13 | -294 | 11 | No | Date of Retraction/Other Unknown; +Notice—Limited or No Information; +Withdrawal; | Unknown | no | no |
| Case 14 | 182 | 172 | No | Concerns/Issues About Data;+Results Not Reproducible;+Unreliable Data; +Unreliable Results; | Falsification and fabrication of data | yes | no |
| Case 15 | 126 | 89 | No | Lack of Approval from Third Party; | Piracy and plagiarism | no | no |
| Case 16 | 86 | 244 | No | Duplication of Image; | Falsification and fabrication of data | no | no |
| Case 17 | 105 | 77 | No | Error by Journal/Publisher;+Retract and Replace; | No misconduct | no | no |
| Case 18 | 255 | 441 | No | Duplication of Image;+Manipulation of Images; | Falsification and fabrication of data | no | no |
| Case 19 | 210 | 993 | No | Error in Analyses;+Unreliable Results; | No misconduct | no | no |
| Case 20 | 99 | 708 | No | Contamination of Materials (General); +Results Not Reproducible; | No misconduct | yes | no |
| Case 21 | 192 | 936 | No | Concerns/Issues About Data; +Concerns/Issues About Image; +Concerns/Issues About Results; +Investigation by Journal/Publisher; +Unreliable Results; | Unknown | no | no |

*(Continued)*

**Table 2.** (Continued)

| | Time from preprint to publication (days) | Time from publication to retraction (days) | Paywalled article | Reason for Retraction | Misconduct classification | Acknowledgment of retraction on preprint | Preprint withdrawn |
|---|---|---|---|---|---|---|---|
| **Case 22** | 468 | 119 | No | Concerns/Issues About Data; +Concerns/Issues About Results; | Falsification and fabrication of data | yes | yes |
| **Case 23** | 60 | 34 | No | Falsification/Fabrication of Image; | Falsification and fabrication of data | yes | no |
| **Case 24** | 38 | 120 | No | +Duplication of Article; | Unknown | no | no |
| **Case 25** | 162 | 418 | No | +Error in Text;+Retract and Replace; | No misconduct | yes | no |
| **Case 26** | 196 | 382 | No | +Error in Analyses; | No misconduct | no | no |
| **Case 27** | 311 | 57 | No | +Contamination of Materials (General); +Results Not Reproducible; | No misconduct | yes | yes |
| **Case 28** | -151 | 209 | No | +Breach of Policy by Author; +Investigation by Company/Institution; +Notice—Limited or No Information; | Piracy and plagiarism | yes | yes |
| **Case 29** | 302 | 91 | No | +Copyright Claims;+Lack of Approval from Third Party; | Piracy and plagiarism | yes | no |
| **Case 30** | 55 | 217 | No | +Breach of Policy by Author;+Conflict of Interest;+Objections by Author(s); +Withdrawal; | Piracy and plagiarism | yes | no |

In 20/30 cases (67%), the retraction was due–at least in part–to some form of research misconduct. Of these, 6 were categorized as Piracy and plagiarism, 10 were categorized as Falsification and fabrication of data, 5 were categorized as Mistreatment of research subjects. In one of these cases, a retraction fell into two categories (Table 2). 8/30 cases (27%) were due to errors, contamination, or irreproducible/unreliable results and did not qualify as research misconduct. In 18/30 cases (60%), the nature of the retraction suggested that the conclusions of the study should no longer be considered valid. In one case, a clear reason for retraction could not be determined, and in another case, the presence or absence of misconduct could not be determined conclusively.

Among the 30 preprints linked to retracted journal publications, 11 (37%) included a clear indication of the retraction on the preprint itself. In 5/30 cases (17%), the preprint itself was marked as withdrawn (Table 2). None of the 30 retracted journal articles visibly indicated the existence of an associated preprint.

## Discussion

Preprints have introduced a new level of speed and transparency to the dissemination of life science research. They have removed barriers to research communication and have particularly benefited early-career researchers, who use them to share their work on their own terms, to show productivity, and to receive valuable feedback from a vast community of peers [12, 13]. However, the rapid growth of preprint servers has also introduced some challenges and complexity into the environment of scholarly publication [14]. The many new preprint servers that have emerged in the past few years have varying budgets, governance, and features as well as disparate policies and operating procedures. The preprint advocacy organization ASAPbio has been pivotal in uniting representatives from the different servers to develop standards and best practices with the aim of establishing consistency in the most important areas, such as

those pertaining to misinformation and trust [15]. Due to various limitations, however, many servers do not have the means and/or the capacity to connect preprints with their associated journal publications.

Journals, for their part, have been generally slow or reluctant to prioritize surfacing preprint links. Only 6% of journals claim to have a mechanism for linking to preprints (transpose-publishing.github.io/#/), yet the actual appearance of such links is much rarer still. Notably, *eLife*, as the first journal to require deposition of a preprint at the point of submission [16], now consistently supplies preprint links on their article pages.

Of note, Google Scholar has its own algorithm for aggregating related publications and their citations and privileging the version of record, even in the absence of formal mechanisms for linking at the preprint server or journal [17]. Over time and with improved sophistication, this technology could address the linkage issues that exist. It already helps to allay concerns over citation dilution–the issue of citations accumulating across different stages and versions of a paper [18]. As the scholarly publishing landscape continues to evolve in the direction of author-led dissemination, enshrining a record of versions is likely to take precedence over the traditional norm of privileging a version of record.

One negative repercussion of the linking gap is that preprints cannot be effectively updated when critical events, such as corrigenda, expressions of concern, or retractions, occur downstream on their journal-published versions. Indeed, the current study shows that even under ideal circumstances–in which links are consistently established by the preprint server–fewer than 50% of preprints indicate a downstream retraction.

Metascience researchers have observed a significant, progressive decline in the time to retraction since 2000 [6]. This increase has been attributed to multiple factors, including wider access to research, which inflates the probability of errors or issues being caught, and increased emphasis on integrity by reputable journals [19]. Articles published in high-impact-factor journals and open access journals would both seem to benefit from better scrutiny–the former potentially being more careful or thorough and the latter being more extensive (by virtue of their broad accessibility). Interestingly, Cokol et al. concluded from their analysis on the burden of retractions that the higher retraction rates observed at high-impact journals are a reflection of the scrutiny their articles receive *post-publication* [20]. Indeed, all 30 of the retractions identified in this study and–as of 2018 –~25% of all retractions in the PubMed database are of open access articles [21].

The time from publication to retraction among the previously preprinted articles in this analysis averaged 9.2 months, notably lower than the average of 33 months observed by Steen et al [6]. Since this discrepancy could simply be attributed to the relatively short time of existence of preprint servers, I limited my analysis of overall time to retraction to only articles published after 2013, when bioRxiv was launched. In this set, the time to retraction averaged 23 months, which is still considerably higher than that of the smaller set of previously preprinted articles. This observation could simply be an artifact of a relatively small sample size, but it might hint at a benefit of early exposure and accessibility.

Due to its integration with Springer Nature journals, Research Square has three (rather than two) mechanisms for linking published papers, so the fidelity of linking is likely to be higher in this preprint server than in others, including bioRxiv and medRxiv. Despite this, a smaller proportion of preprints are linked to journal publications on Research Square compared to bioRxiv and medRxiv. There are a number of factors that could account for this discrepancy, including known technical deficits preventing reliable linking of Research Square articles to Springer Nature submissions, the longer time of existence of bioRxiv and medRxiv relative to Research Square, the multidisciplinary nature of the Research Square server,

differential screening procedures between the servers, and the quality of preprints that the servers receive.

In this study, fewer than half of the preprints were clearly marked with an indication of the downstream retraction. Preprint servers that issue DOIs via Crossref have the ability to use the "is-preprint-of" metadata relationship to link preprints to their downstream publications. This makes it easier to check for updates in the Crossref metadata associated with the journal publication. However, this requires that journals properly register retractions and other events via Crossref and that preprint servers both initiate the link and regularly check it against Crossref's metadata. Crossmark–the Crossref service that provides public-facing updates on post-publication events–is not currently enabled on preprint platforms, so the platforms must establish their own mechanisms for finding and surfacing this information. Across the rapidly growing landscape of preprint servers and journals that currently exist, it is unlikely that this occurs reliably. This failure to back-propagate critical information not only leaves preprint readers in the dark about the invalidation of some research, but it could also exacerbate the problem of papers being cited persistently after retraction [22, 23]. To be clear, the problem is not with the occurrence of retractions themselves–which should be viewed as an indication that corrective systems are working properly [24]–but, rather, with the persistence of these papers in the literature due to their continued citation.

In 5 out of the 30 retractions identified in this study, the preprint had also been marked as withdrawn. Withdrawal is considered analogous to retraction in the preprint sphere [25], but the question of whether a preprint should be withdrawn following retraction of a later version has not been addressed in any formal capacity. ASAPbio, an organization promoting the productive use of preprints as tools for research dissemination, currently does not include downstream retraction as cause for withdrawal in their published recommendations for preprint withdrawal and removal, indicating that preprint servers should be notified and left to decide the appropriate course of action in each individual case. These guidelines also emphasize that while journals should make an effort to alert preprint servers to these events, it is ultimately the author's responsibility to update both parties about actions taken on an associated record [25]. However, as it is not uncommon for authors to disagree with a retraction or become unresponsive to inquiries about issues with their publications, it may be unrealistic to rely on authors to properly propagate such information.

Importantly, even if automated connections via Crossref and Crossmark are established for all preprint servers, several issues will persist. First, journal-published versions whose titles and author lists do not align with their preprints will fail to be linked. Second, only updates on retractions, which are issued their own DOIs, will be facilitated. Unless journals take responsibility for establishing connections to previously published versions, linkage will continue to be suboptimal and preprint readers will continue to be oblivious to events such as expressions of concern, which can take months or even years of investigation before resolving or resulting in a retraction [26].

Preprints have proven valuable to researchers [13, 18] and are likely to become a fixture among authors of biological and medical research, increasingly becoming the earliest version of a study that is shared with the world. But as preprints become more common, so too will the incidence of downstream retractions or other problems that are not properly accounted for on the preprint. As adoption of preprints continues to grow, serious consideration should be given to ensuring that preprints are digitally connected with associated publications and building reliable mechanisms for propagating critical updates. Future studies should include an analysis of preprints and their journal article links across the broader group of preprint servers to provide a more comprehensive picture of the state of information propagation across the publication continuum.

### Limitations

Counts of linked articles via Crossref are known to be limited to near exact matches of titles and author lists between preprints and journal publications. For Research Square, counts of links established using internal mechanisms are also underestimates due to ongoing technical deficits that prevent perfect linking of Research Square preprints to Springer Nature articles. Thus, the actual percentages of preprints that are later published is likely higher than represented by the counts presented here.

### Acknowledgments

My sincere thanks to Ivan Oransky and Alison Abritis of Retraction Watch for their guidance and access to the Retraction Watch database.

### Author Contributions

**Conceptualization:** Michele Avissar-Whiting.

**Data curation:** Michele Avissar-Whiting.

**Formal analysis:** Michele Avissar-Whiting.

**Investigation:** Michele Avissar-Whiting.

**Methodology:** Michele Avissar-Whiting.

**Project administration:** Michele Avissar-Whiting.

**Software:** Michele Avissar-Whiting.

**Visualization:** Michele Avissar-Whiting.

**Writing – original draft:** Michele Avissar-Whiting.

**Writing – review & editing:** Michele Avissar-Whiting.

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
