## [Decision Letter · Decision Letter 0]

16 Mar 2022

PONE-D-22-05282Downstream retraction of preprinted research in the life and medical sciencesPLOS ONE

Dear Dr. Avissar-Whiting,

Thank you for submitting your manuscript to PLOS ONE. After careful consideration, we feel that it has merit but does not fully meet PLOS ONE’s publication criteria as it currently stands. Therefore, we invite you to submit a revised version of the manuscript that addresses the points raised during the review process. The manuscript lacks referencing to related work. This topic has been and is discussed in a number of articles that are not sufficiently cited.

We look forward to receiving your revised manuscript.

Kind regards,

Frederique Lisacek

Academic Editor

PLOS ONE

Journal Requirements:

"Michele Avissar-Whiting is the Editor in Chief of the Research Square preprint platform"

Reviewers' comments:

Reviewer's Responses to Questions

**Comments to the Author**

1. Is the manuscript technically sound, and do the data support the conclusions?

Reviewer #1: Yes

Reviewer #2: Yes

2. Has the statistical analysis been performed appropriately and rigorously? 

Reviewer #1: Yes

Reviewer #2: N/A

3. Have the authors made all data underlying the findings in their manuscript fully available?

Reviewer #1: Yes

Reviewer #2: Yes

4. Is the manuscript presented in an intelligible fashion and written in standard English?

Reviewer #1: Yes

Reviewer #2: Yes

5. Review Comments to the Author

Reviewer #1: In my opinion this paper is relevant because it gives light on the link between preprints, publications focusing on retracted papers.

It is very relevant to know how journals manage retraction but also how preprint servers identify those retracted papers from journals.

This paper addresses this questions and provide some responses.

I have very few comments.

The first one is related to the references. In general there is a very low number of references and a small effort should be done by the author on this. In fact, the introduction only has 6 references with some paragraphs with no reference

I have some doubts on the classification on retracted papers, from table 1 you can see that a number of retractions (most of them) were due to misconduct, but some of them did not show misconduct. This is the case of case 17, where it was a journal mistake. Regarding IRB approval, though it is bad practice, it is not misconduct as such on how we understand publishing-related misconduct. I wonder if you could be more precise on this in your results and how you consider this case 17.

From table 1. It should be included the journal and name of first author. I think that journals publishing retracted papers should be identified along with unethical authors.

Methods. Second paragraph. A reference is needed here, because you are classifying misconduct. It this your own classification? There are other classifications published that could add more comparability to your results.

Results. Line 122-126. The cases do not add 30, and they should. I guess that perhaps some retractions are classified in two categories. An explanation is needed here.

You give importance on the number of retractions by preprint server. I suggest that a new table showing retractions per server and also retraction notice per server, along with time to notice or time to retraction would provide useful information on how these servers work when retractions are detected and their performance on this.

Some more references are needed on the discussion.

Professor Alberto Ruano-Ravina

Reviewer #2: This work provides us with a convincing observation on the status of preprints and links between preprints and subsequent pubications. This is rather an opinion on a socio-political topic of great interest than an authentic research article, but it is worth communicating to the community

6. PLOS authors have the option to publish the peer review history of their article (what does this mean?). If published, this will include your full peer review and any attached files.

Reviewer #1: **Yes: **Alberto Ruano-Ravina

Reviewer #2: **Yes: **Antoine Danchin

---

## [Author Response · Author response to Decision Letter 0]

21 Mar 2022

Dear Dr. Lisacek,

I am very grateful to the referees for the time they invested in reviewing the manuscript and for their important feedback, which has led to important revisions that greatly improved the presentation. My point-by-point responses to the referees’ comments follow.

---

Comments to the Author

Reviewer #1: In my opinion this paper is relevant because it gives light on the link between preprints, publications focusing on retracted papers.

It is very relevant to know how journals manage retraction but also how preprint servers identify those retracted papers from journals.

This paper addresses this questions and provide some responses.

I have very few comments.

The first one is related to the references. In general there is a very low number of references and a small effort should be done by the author on this. In fact, the introduction only has 6 references with some paragraphs with no reference.

Response: I agree that the manuscript was lacking sufficient references. I have now added six new relevant references, including in places where new information was added based on other suggestions by the reviewer. I have also added new citations to existing references where needed throughout the text.

---

I have some doubts on the classification on retracted papers, from table 1 you can see that a number of retractions (most of them) were due to misconduct, but some of them did not show misconduct. This is the case of case 17, where it was a journal mistake. Regarding IRB approval, though it is bad practice, it is not misconduct as such on how we understand publishing-related misconduct. I wonder if you could be more precise on this in your results and how you consider this case 17.

Response: While not all cases were defined as being due to misconduct in the original manuscript, the reviewer makes an excellent point that these categories should have been defined in a more standardized and clearer way. To clarify the categorization of these retractions, I have redesignated them based on the definitions of research misconduct provided by the Council of Science Editors. Three discrete designations have been used, and the following text has been added to the Methods section:

“Misconduct in each case of retraction was categorized according to the areas defined by the Council of Science Editors [7] as follows: Mistreatment of research subjects (includes failure to obtain approval from an ethical review board or consent from human subjects before conducting the study and failure to follow an approved protocol); Falsification and fabrication of data; Piracy and plagiarism (including unauthorized use of third-party data or other breach of policy by authors); or No evidence of misconduct. The determination of presence and type of misconduct was based on information contained in the individual retraction notices as well as on the reasons for retraction briefly noted in the Retraction Watch database.” 

Note that this definitional change also changed the quantification of retractions in each category, so the numbers in the results have shifted slightly as a result.

---

From table 1. It should be included the journal and name of first author. I think that journals publishing retracted papers should be identified along with unethical authors.

Response: While I agree in principle with the referee’s comment, I could not include this information due to our agreement with the Center for Scientific Integrity, which limits the granularity of the data I could share in this publication. Additionally (and fortunately), the specifics of the individual studies are not relevant to the thesis of this study, which is focused more on the integrity of information propagation rather than the individual instances of retraction themselves. Thus, not identifying them here does not seem to undermine the fundamental premise of the work. On a related note, I have removed the Article Type and Country information from the table, as neither is discussed or relevant to the topic at hand.

---

Methods. Second paragraph. A reference is needed here, because you are classifying misconduct. It this your own classification? There are other classifications published that could add more comparability to your results.

Response: Thank you for this comment, which has now been addressed above.

---

Results. Line 122-126. The cases do not add 30, and they should. I guess that perhaps some retractions are classified in two categories. An explanation is needed here.

Response: Thank you for this comment. The reviewer is correct that retractions could fall into multiple categories. I have now acknowledged this clearly in this section. Note that substantial changes were made to this section due to the re-classification of research misconduct.

---

You give importance on the number of retractions by preprint server. I suggest that a new table showing retractions per server and also retraction notice per server, along with time to notice or time to retraction would provide useful information on how these servers work when retractions are detected and their performance on this.

Response: I’m grateful to the reviewer for this suggestion. I have added a table (new Table 1) that contains the server-specific information. I have left the retraction notice information in the larger table, however.

---

Some more references are needed on the discussion.

Response: I thank the reviewer for this suggestion. Six new references and four new citations to existing references have been added to the paper.

---

Reviewer #2: This work provides us with a convincing observation on the status of preprints and links between preprints and subsequent pubications. This is rather an opinion on a socio-political topic of great interest than an authentic research article, but it is worth communicating to the community

Response: I thank the reviewer for their comment and agree that there is certainly an element of opinion or “call to action” in this work. My hope is that I’ve provided sufficient empirical evidence to start a discussion and provide a foundation for future, more comprehensive, studies assessing the fidelity of information transfer between associated article types. More importantly, I hope it will convince the myriad stakeholders of the importance of strengthening these connections. This comment compelled me to add a statement regarding future studies to the end of the discussion.

---

## [Decision Letter · Decision Letter 1]

20 Apr 2022

Downstream retraction of preprinted research in the life and medical sciences

PONE-D-22-05282R1

Dear Dr. Avissar-Whiting,

We’re pleased to inform you that your manuscript has been judged scientifically suitable for publication and will be formally accepted for publication once it meets all outstanding technical requirements.

Kind regards,

Frederique Lisacek

Academic Editor

PLOS ONE

Additional Editor Comments (optional):

Reviewers' comments:

Reviewer's Responses to Questions

**Comments to the Author**

1. If the authors have adequately addressed your comments raised in a previous round of review and you feel that this manuscript is now acceptable for publication, you may indicate that here to bypass the “Comments to the Author” section, enter your conflict of interest statement in the “Confidential to Editor” section, and submit your "Accept" recommendation.

Reviewer #1: All comments have been addressed

2. Is the manuscript technically sound, and do the data support the conclusions?

Reviewer #1: Yes

3. Has the statistical analysis been performed appropriately and rigorously? 

Reviewer #1: Yes

4. Have the authors made all data underlying the findings in their manuscript fully available?

Reviewer #1: Yes

5. Is the manuscript presented in an intelligible fashion and written in standard English?

Reviewer #1: Yes

6. Review Comments to the Author

Reviewer #1: The author has answered all my comments satisfactorily and I do not have more concerns on the contents of the manuscript

7. PLOS authors have the option to publish the peer review history of their article (what does this mean?). If published, this will include your full peer review and any attached files.

Reviewer #1: **Yes: **Alberto Ruano-Raviña

---

## [Editor Report · Acceptance letter]

22 Apr 2022

PONE-D-22-05282R1 

Downstream retraction of preprinted research in the life and medical sciences 

Dear Dr. Avissar-Whiting:

I'm pleased to inform you that your manuscript has been deemed suitable for publication in PLOS ONE. Congratulations! Your manuscript is now with our production department. 

Kind regards, 

on behalf of

Dr. Frederique Lisacek 

Academic Editor

PLOS ONE